# Unnatural Languages Are Not Bugs but Features for LLMs

**Keyu Duan**[1][*]   **Yiran Zhao**[1][*]   **Zhili Feng**[2]   **Jinjie Ni**[1]   **Tianyu Pang**[3]   **Qian Liu** [3]
**Tianle Cai**[4]   **Longxu Dou**[3]   **Kenji Kawaguchi**[1]   **Anirudh Goyal**[5]   **J. Zico Kolter**[2]
**Michael Qizhe Shieh**[1][†]
[1] National University of Singapore   [2] Carnegie Mellon University   [3] Sea AI Lab
[4] Princeton University   [5] Mila, University of Montreal

## Abstract

Large Language Models (LLMs) have been observed to process non-human-readable text sequences, such as jailbreak prompts, often viewed as a bug for aligned LLMs. In this work, we present a systematic investigation challenging this perception, demonstrating that unnatural languages - strings that appear incomprehensible to humans but maintain semantic meanings for LLMs - contain latent features usable by models. Notably, unnatural languages possess latent features that can be generalized across different models and tasks during inference. Furthermore, models fine-tuned on unnatural versions of instruction datasets perform on-par with those trained on natural language, achieving $49.71$ win rates in Length-controlled AlpacaEval 2.0 in average across various base models. In addition, through comprehensive analysis, we demonstrate that LLMs process unnatural languages by filtering noise and inferring contextual meaning from filtered words.

## 1 Introduction

Large Language Models (LLMs) (OpenAI, 2023; Touvron et al., 2023; Dubey et al., 2024; anthropic, 2024) have shown remarkable capabilities in understanding and generating human-readable text, achieving impressive performance across tasks, spanning from question answering (Bisk et al., 2020; Ni et al., 2024) and mathematical reasoning Cobbe et al. (2021); Hendrycks et al. (2021); Gao et al. (2024) to open-ended dialogue (Li et al., 2023). Such abilities are largely attributed to targeted alignment training (Wei et al., 2021; Ouyang et al., 2022), which post-train models to better follow instructions and adhere to preferred behaviors.

Despite being specifically tuned, non human-readable data sometimes can unexpectedly influence model behavior. In computer vision, Ilyas et al. (2019); Nguyen et al. (2015) find that seemingly unrecognizable images could be leveraged to train reasonable good image classification models. This phenomenon extends to natural language processing, where Zou et al. (2023) demonstrate that LLMs could also be prompted with an unreadable suffix to generate objectionable outputs, even though LLMs are well-trained for not doing so. Besides, Pfau et al. (2024) discovers that by appending non human-readable filler tokens to the input, LLMs could solve algorithmic tasks more accurately. However, regarding LLMs' surprising behaviors in response to non human-readable inputs, there lacks systematic studies exploring the properties and applications of such non human-readable strings and interpreting the underlying mechanisms in LLMs. This raises a fundamental question: *whether these non human-readable strings are truly devoid of meaning or contain latent features usable by models?*

To answer the above research questions, we study a phenomenon named *unnatural languages* - strings that deviate from natural language syntax and appear extremely noisy to human readers, yet remain understandable to LLMs. Specifically, as illustrated in Figure 1(a), we propose an approach to search for a semantically equivalent but syntactically unnatural version of the natural string, where

---

[*]Equal contribution. Keyu Duan is the project leader. Yiran Zhao is the senior student contributor
[†]Correspondence to: Keyu Duan (k.duan@u.nus.edu), Yiran Zhao (zhaoyiran@u.nus.edu), Michael Qizhe Shieh (michaelshieh@comp.nus.edu.sg).

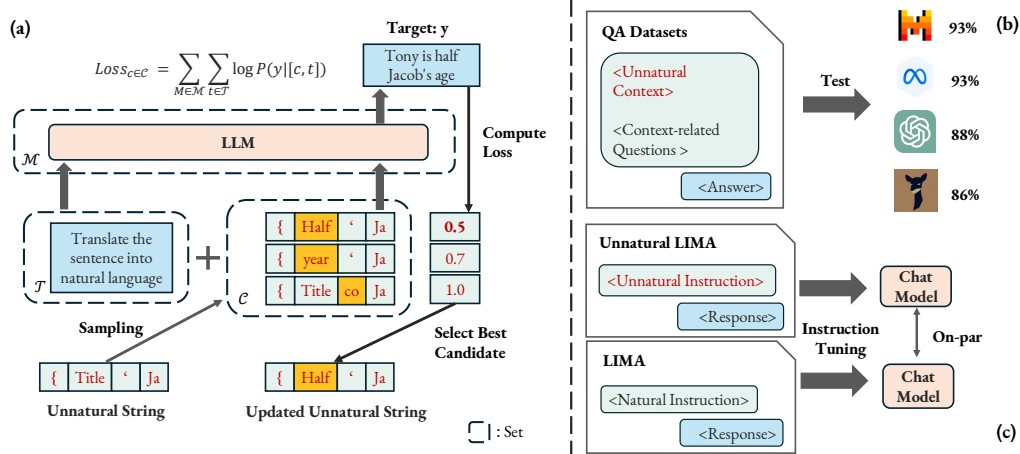

Figure 1: *(a)*: Unnatural languages searching method. *(b)*: We construct question-answering tasks using unnatural contexts and discover that unnatural languages can be directly transferred across a broader range of tasks and understood by diverse LLMs. (c): LLMs fine-tuned on unnatural versions of instruction datasets perform on-par with those trained on natural language.

semantic equivalence is established through their ability to be translated back to natural form via models performing translation inference. For the searching process, we employ a gradient-based stochastic sampling procedure to obtain a set of candidate unnatural strings; evaluate their probability of being translated back to the corresponding natural versions across multiple models; and select the unnatural candidate that yields the highest probability. We repeat the process until either convergence occurs or a maximum number of iterations is reached, with the final converged string representing the semantically equivalent unnatural string.

We first explore whether these unnatural languages possess latent features that can be generalized across different models and tasks during inference. To investigate this, as shown in Figure 1(b), we construct several context-based question-answering datasets, where the context is unnatural obtained by the searching approach while natural questions related to the context are provided. Specifically, to prevent models from relying on common-sense memory when answering questions without context, we develop SynContextQA, a synthetic dataset generated by another LLM, containing contexts about non-existent entities paired with corresponding questions. We then transformed natural contexts into unnatural ver-

Table 1: Concrete examples of natural contexts' unnatural version in SycContextQA and SimGSM8K.

| Natural Context | Unnatural Version |
|---|---|
| The stock price of Gold-Mine Inc. increased by 20% last week. | (alt+eqn={\\>; {};The\\,\\stock baaelkrie@nuier priceungeureau got sich last '#GM;;heidisation Inc. weekestig %}20% durch'),png encrypt render \"OK Gold-Mine.",preventDefault |
| Carly collected 7 starfish with 5 arms each and one seastar with 14 arms. | \|Each and : algebra dinner! absolutely 7 do): shortly . seastar collectedthe '' kW)\$, one !5 ! 14' starfish with sic}}_{\label Carly} arms. Onehorailey constructed WriteStatus(\$\$\Toggle Zwezeich-nung OK |

sions using the unnatural language searching method while preserving the original questions. Additionally, to ensure models do not simply extract keywords from unnatural contexts in SynContextQA, we create SimGSM8K, a dataset of simple questions derived from GSM8K (Cobbe et al., 2021). We chose simple questions to minimize the impact of reasoning ability on our results and focus primarily on unnatural language comprehension. As with SynContextQA, we transformed these contexts into unnatural versions. Table 1 shows concrete examples of two datasets. With these datasets, we test a large variety of LLMs, including open-source models as well as commercial models. The results show that compared to natural context, all models can recover $82.0\%$ of the original accuracy on our constructed SynContextQA dataset and $61.6\%$ on SimGSM8K, demonstrating that unnatural languages contain latent features that enable comprehension across different scenarios.

Moreover, we explore whether these unnatural languages possess transferable latent features that can be effectively utilized in instruction tuning to improve models' instruction-following capabilities.

Specifically, as shown in Figure 1(c), we employ a high-quality but small size instruction tuning dataset LIMA (Zhou et al., 2023), and we replace the original instructions with our equivalent unnatural versions searched using our proposed approach. We show that the models fine-tuned on it and the original have on-par performance on prestigious benchmarks, including Length-controlled (LC) AlpacaEval 2.0 (Li et al., 2023) and MixEval (Ni et al., 2024). Particularly, on LC AlpacaEval 2.0, the three models — Llama-3-8B (Dubey et al., 2024), Gemma-2-9B Team et al. (2024), and Llama-3-70B (Dubey et al., 2024) — tuned on the unnatural LIMA achieves an winrate of $49.78\%$, $47.13\%$, and $52.22\%$ against the corresponding models tuned on the natural LIMA, respectively.

These findings strongly demonstrate our key findings: unnatural languages are not bugs but features for LLMs. In addition, we attempt to understand the mechanisms by which LLMs process such unnatural languages. We demonstrate that LLMs process unnatural languages by effectively filtering out irrelevant tokens. Furthermore, LLMs combine relevant tokens from unnatural languages and infer contextual meaning in response to natural version questions.

## 2 UNNATURAL LANGUAGES SEARCHING METHOD

In this section, we introduce our approach for searching the unnatural version of given natural string.

**Problem description.** We denote a natural string as $S$ and its equivalent unnatural version as $S'$. The equivalence between them is formally defined as $S \equiv \mathcal{LLM}_M(S'|t)$, where $t$ represents a reconstruction task—such as translating the unnatural sentence into natural language—and $\mathcal{LLM}_M(S'|t)$ denotes the output of model $M$ given input $S'$ under task prompt $t$. Furthermore, we define the log-probability of model $M$ generating natural string $S$ when given unnatural string $x$ under task prompt $t$ as $\log P_M(S|x,t)$. Therefore, the unnatural string searching problem can be formulated as

$$S' := \arg\max_{x \in \mathcal{X}} \log P_M(S|x,t), \tag{1}$$

where $\mathcal{X}$ represents the unnatural languages space, encompassing all possible strings of arbitrary length. However, searching the entire unnatural space is computationally infeasible. For simplicity and without loss of generality, we constrain $x$ to maintain a fixed length at the token level, i.e., $x \in \mathcal{X}_M$, where $\mathcal{X}_M \triangleq \{x \,||\, \text{tokenize}_M(x)| = n, x \in \mathcal{X}\}$ and $n$ is a predefined constant length.

Furthermore, to enhance the generalizability of the obtained unnatural languages, we employ multiple models, denoted as $\mathcal{M} = \{M_1, M_2, \ldots, M_k\}$, to collaboratively search for unnatural strings. Additionally, we introduce a set of tasks $\mathcal{T} = \{t_1, t_2, \ldots, t_m\}$ as a collaborative optimization objective. Therefore, our goal of searching the equivalent unnatural string $S'$ is formulated as solving the following optimization problem:

$$S' := \arg\max_{x \in \bigcup_{M \in \mathcal{M}} \mathcal{X}_M} \sum_{M \in \mathcal{M}} \sum_{t \in \mathcal{T}} \log P_M(S|x,t). \tag{2}$$

**Algorithm description.** The optimization problem defined in Equation 2 is a discrete optimization problem as $\mathcal{X}_M$ is a discrete space of size $|\mathcal{V}_M|^n$, where $\mathcal{V}_M$ denotes the vocabulary set of the model $M$. Due to the discrete nature of the problem, gradient-based optimization methods cannot be directly applied. Furthermore, the search space is too vast for exhaustive exploration. Therefore, we propose a sample-and-selection algorithm inspired by the optimization approaches of Shin et al. (2020); Zou et al. (2023). Specifically, in each optimization iteration, the unnatural string $x$ is first tokenized by model $M$ into $\mathbf{x}_{1:n}$. For each position, we identify the top-k most influential tokens $\mathbf{X}_{1:n}$ based on the gradient of the optimization objective of Equation 2, i.e.,

$$\text{Top-k}\left(\nabla_{\mathbf{x}_{1:n}} \sum_{t \in \mathcal{T}} \log P_M(S|\mathbf{x}_{1:n}, t)\right). \tag{3}$$

We then generate $B$ candidates $\{\tilde{\mathbf{x}}_{1:n}^{(1)}, \tilde{\mathbf{x}}_{1:n}^{(2)}, \cdots, \tilde{\mathbf{x}}_{1:n}^{(B)}\}$, where each candidate differs from $\mathbf{x}_{1:n}$ by exactly one token, randomly sampled from $\mathbf{X}_{1:n}$. These candidates are then decoded back to strings $\{\tilde{x}^{(1)}, \tilde{x}^{(2)}, \cdots, \tilde{x}^{(B)}\}$ by model $M$ for subsequent cross-model unification optimization. This candidate generation process is applied across all models in $\mathcal{M}$, yielding $B|M|$ total candidates. The candidate with the optimal loss is selected for the next iteration. This process continues until convergence or until reaching a pre-defined number of iterations.

Table 2: Performance comparison for different contexts across different models on SynContextQA and SimGSM8K datasets. All answers were generated under zero-shot setting without sampling. "Direct" refers to models used for unnatural languages searching, while "Transfer" indicates the implementation of searched unnatural languages.

| | Model | SynContextQA | | | SimGSM8K | | |
|---|---|---|---|---|---|---|---|
| | | Natural | Shuf-InJ | Unnatural | Natural | Shuf-InJ | Unnatural |
| **Direct** | Mistral-7B-Instruct-v0.1 | 0.89 | 0.55 | 0.93 | 0.85 | 0.20 | 0.42 |
| | Vicuna-7B-v1.5 | 0.96 | 0.40 | 0.86 | 0.63 | 0.12 | 0.20 |
| | *Average* | 0.93 | 0.48 | **0.90** | 0.74 | 0.16 | **0.31** |
| **Transfer** | Meta-Llama-3-8B-Instruct | 0.99 | 0.29 | 0.63 | 0.58 | 0.18 | 0.50 |
| | Gemma-2-9B-Instruct | 0.98 | 0.35 | 0.65 | 0.97 | 0.21 | 0.41 |
| | Meta-Llama-3-70B-Instruct | 0.97 | 0.73 | 0.93 | 1.00 | 0.38 | 0.75 |
| | GPT-3.5-turbo | 0.98 | 0.73 | 0.93 | 0.91 | 0.32 | 0.53 |
| | GPT-4o | 0.98 | 0.61 | 0.88 | 0.95 | 0.25 | 0.53 |
| | *Average* | 0.98 | 0.54 | **0.80** | 0.88 | 0.27 | **0.54** |

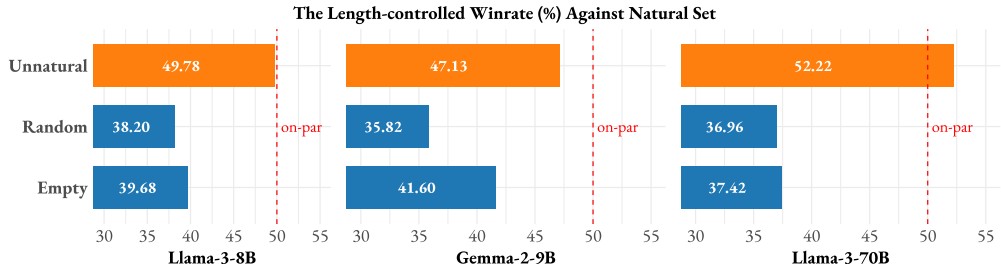

Figure 2: The winrate (%) against natural set on LC Alpaca 2.0 for various models. A winrate of 50% indicates on-par performance.

**Implementation details.** In practice, tokenized sequences of $x$ have varying lengths across optimization iterations due to different models employing distinct tokenizers, with no direct one-to-one mapping between tokens and words. To maintain generalizability and in line with Zou et al. (2023); Zhao et al. (2024), we initialize $x$ through a combination of shuffling words in $S$ and randomly inserting several special characters "!". Algorithm 1 in Appendix A1 provides a detailed illustration of the searching algorithm. In addition, we conduct verification experiments to demonstrate that our searched unnatural strings can be accurately translated back to their natural versions. Details are further illustrated in Appendix A2.

# 3 UNNATURAL LANGUAGES CAN BE UNDERSTOOD ACROSS TASKS AND LLMS

In this section, we investigate whether unnatural languages—generated by Algorithm 1 via reconstruction tasks across multiple models—can be directly transferred across a broader range of tasks and understood by diverse LLMs.

## 3.1 EXPERIMENT SETUP

To evaluate LLMs' genuine understanding of unnatural languages, we design questions closely related to unnatural contexts. We employ context-based question-answering problems where the context is expressed in unnatural languages while maintaining questions in natural language. This approach helps isolate the models' understanding of unnatural languages without introducing additional comprehension challenges.

**Benchmarks.** ❶ *SynContextQA*. We begin with a classic commonsense question-answering task, where questions are asked in relation to given contextual knowledge. However, using existing commonsense QA datasets presents a challenge, as LLMs may answer questions based on their pre-trained knowledge rather than the provided context. To address this, we leverage GPT-3.5 (Achiam

Table 3: The results of model variants instruction tuned on different types of LIMA (including natural, unnatural, random, and empty instruction (Hewitt et al., 2024)) on MixEval (Ni et al., 2024). Each dataset represents the subsets selected by MixEval based on the real-world data distribution. M.C. and F.F. denote multiple-choice and free-form, respectively. Particularly, we remove the results of subset GPQA, MBPP, WinoGrande, and HumanEval since the subsets were too small (less than ten test cases) for consistent evaluation.

| Type | Dataset | Llama-3-8B | | | | Gemma-2-9B | | | | Llama-3-70B | | | |
|---|---|---|---|---|---|---|---|---|---|---|---|---|---|
| | | Natural | Random | Empty | Unnatural | Natural | Random | Empty | Unnatural | Natural | Random | Empty | Unnatural |
| M.C | ComsenseQA | 0.530 | 0.569 | 0.604 | 0.579 | 0.599 | 0.550 | 0.495 | 0.668 | 0.748 | 0.668 | 0.559 | 0.693 |
| | BoolQ | 0.614 | 0.649 | 0.673 | 0.678 | 0.567 | 0.614 | 0.632 | 0.673 | 0.830 | 0.708 | 0.632 | 0.848 |
| | OpenBookQA | 0.581 | 0.628 | 0.721 | 0.721 | 0.605 | 0.744 | 0.744 | 0.721 | 0.814 | 0.744 | 0.721 | 0.837 |
| | SIQA | 0.462 | 0.570 | 0.624 | 0.462 | 0.613 | 0.516 | 0.645 | 0.538 | 0.742 | 0.581 | 0.495 | 0.677 |
| | HellaSwag | 0.338 | 0.364 | 0.360 | 0.328 | 0.331 | 0.344 | 0.289 | 0.357 | 0.461 | 0.373 | 0.351 | 0.364 |
| | MMLU-Pro | 0.357 | 0.346 | 0.308 | 0.335 | 0.427 | 0.438 | 0.459 | 0.432 | 0.503 | 0.427 | 0.465 | 0.578 |
| | AGIEval | 0.331 | 0.359 | 0.340 | 0.352 | 0.370 | 0.314 | 0.407 | 0.349 | 0.566 | 0.423 | 0.426 | 0.543 |
| | PIQA | 0.514 | 0.676 | 0.705 | 0.600 | 0.781 | 0.695 | 0.638 | 0.733 | 0.790 | 0.752 | 0.724 | 0.886 |
| | MMLU | 0.658 | 0.634 | 0.661 | 0.633 | 0.724 | 0.680 | 0.700 | 0.718 | 0.811 | 0.736 | 0.731 | 0.805 |
| | ARC | 0.802 | 0.802 | 0.780 | 0.791 | 0.923 | 0.901 | 0.879 | 0.923 | 0.956 | 0.835 | 0.868 | 0.934 |
| | *Average* | 0.545 | 0.563 | **0.579** | 0.552 | 0.607 | 0.585 | 0.582 | **0.623** | 0.721 | 0.635 | 0.611 | **0.707** |
| F.F. | TriviaQA | 0.591 | 0.453 | 0.452 | 0.558 | 0.609 | 0.481 | 0.563 | 0.585 | 0.829 | 0.638 | 0.685 | 0.825 |
| | BBH | 0.537 | 0.633 | 0.526 | 0.606 | 0.621 | 0.438 | 0.700 | 0.687 | 0.817 | 0.670 | 0.484 | 0.693 |
| | DROP | 0.584 | 0.385 | 0.484 | 0.545 | 0.638 | 0.481 | 0.638 | 0.651 | 0.755 | 0.585 | 0.631 | 0.767 |
| | MATH | 0.381 | 0.290 | 0.510 | 0.394 | 0.490 | 0.668 | 0.568 | 0.410 | 0.668 | 0.642 | 0.606 | 0.610 |
| | GSM8K | 0.593 | 0.470 | 0.535 | 0.500 | 0.675 | 0.460 | 0.715 | 0.545 | 0.873 | 0.827 | 0.817 | 0.787 |
| | *Average* | 0.583 | 0.445 | 0.467 | **0.554** | 0.616 | 0.481 | 0.592 | **0.603** | 0.809 | 0.631 | 0.662 | **0.799** |
| **Overall Average** | | 0.557 | 0.499 | 0.516 | **0.547** | 0.605 | 0.526 | 0.582 | **0.605** | 0.760 | 0.627 | 0.631 | **0.748** |

et al., 2023) to generate knowledge about non-existing entities and their corresponding questions, named as SynContextQA, thereby ensuring the model must derive answers from the given context rather than rely on pre-existing information. Prompts for generation and post-processing are illustrated in Appendix A3. We then transformed natural contexts into unnatural versions using the unnatural languages searching method while preserving the original questions. ❷ *SimGSM8K*. Furthermore, to ensure models do not simply extract keywords from unnatural contexts in SynContextQA, we test the unnatural languages on GSM8K (Cobbe et al., 2021), a more complex task requiring reasoning capability. As our primary objective is to assess the ability to comprehend unnatural languages, rather than to evaluate reasoning ability, we select 100 relatively simple questions from the whole test set, named SimGSM8K. We show a concrete example for each dataset in Table 1.

**Backbone Models.** We select a diverse range of LLMs, spanning from smaller open-source models to larger closed-source ones. Specifically, we use Mistral-7B-Instruct-v0.1 (Jiang et al., 2023), Vicuna-7B-v1.5 (Chiang et al., 2023), Meta-Llama-3-8B-Instruct (Dubey et al., 2024), Gemma2-9B-Instruct (Team et al., 2024), Meta-Llama-3-70B-Instruct, GPT-3.5-turbo (Achiam et al., 2023), GPT-4o (Hurst et al., 2024). Furthermore, to balance efficiency in the unnatural languages searching algorithm with generalizability, we employ Mistral-7B-Instruct-v0.1 and Vicuna-7B-v1.5, two renowned open-source models from distinct series, as the model set $M$ in Algorithm 1.

**Baselines.** We mainly employ two baselines. (i) natural language, which uses the original unmodified text, and (ii) shuffled language with injected special tokens (Shuf-Inj), which serves as the initialization step for our unnatural languages search algorithm in Algorithm 1.

**Experiment Details.** For SynContextQA, we evaluate performance using exact keyword matching, while for SimGSM8K, we use accuracy as the evaluation metric.

## 3.2 MAIN RESULTS

As shown in Table 2, for SynContextQA, the test accuracy of all models in unnatural languages is on-par with the one of natural language, by a large margin with Shuf-InJ. In average, the test accuracy of transferred models in unnatural languages is 80.4%. For SimGSM8K, the test accuracy of most models on natural questions is over 80%, indicating the simplicity of questions, thus mitigating the concerning of question complexity. Meanwhile, the performance of close-source models could also answer half of the questions correctly, outperforming the Shuf-InJ by an averaged margin of

26.7%. Both results indicate that such unnatural languages is highly transferrable across models with different architectures and training corpus, including GPT-4o (Hurst et al., 2024), which is considered as the most well-aligned models. As a result, it mitigates the conjecture that such unnatural languages is a glitch of specific LLMs, but a general phenomenon and inherent property for LLMs.

It is worthwhile to note that there is a significant performance gap between the natural and unnatural set for SimGSM8K. This gap can be attributed to the increased complexity of SimGSM8K contexts, which typically comprise multiple interconnected sentences. Furthermore, SimGSM8K questions require sophisticated multi-step reasoning processes, making them substantially more challenging than standard SynContextQA tasks. However, this is not the upper bound of the performance of LLMs on unnatural languages, once the unnatural languages searching approach could be further improved.

In addition, we extend our investigation into the understanding of unnatural languages in a dialogue format, which serves as the foundation for LLM agents. Further details can be found in Appendix A6.

## 3.3 Further Analysis

To broaden the scope of the unnatural languages, we implemented it in the base model rather than only the chat version, demonstrating that models can truly understand these unnatural languages without relying on chat models' ability to understand noisy instructions.

**Experiment Settings.** For the base model, we employ in-context learning (ICL) with examples in natural language to ensure consistent output formatting while avoiding unnatural languages patterns in the learning process.

Table 4: The performance comparison of different pre-trained base models on SimGSM8K. Ratio is calculated as the performance on unnatural languages divided by the performance on natural ones.

| Model | Prompt | Natural | Unnatural | Ratio |
|---|---|---|---|---|
| Mistral-7B | ICL (n=1) | 0.70 | 0.23 | 0.33 |
| | ICL (n=8) | 0.71 | 0.38 | 0.53 |
| Llama-3-8B | ICL (n=1) | 0.74 | 0.33 | 0.45 |
| | ICL (n=8) | 0.87 | 0.42 | 0.48 |

**Main Results.** As shown in Table 4, under in-context learning setting with 8 examples, the unnatural test accuracy of pre-trained base models before alignment achieves 38% and 42% in average, respectively. Particularly, considering the success ratio (i.e. unnatural acc./natural acc.), the ratio achieves 53% and 48%, respectively. This indicates that pre-trained model could inherently understand the unnatural languages without alignment.

## 4 LLMs Can Learn Instruction Following Capabilities From Unnatural Languages

In this section, we explore the properties of unnatural languages from the perspective of post-training.

## 4.1 Experiment Setup

We explore whether the instruction fine-tuning pre-trained LLMs on unnatural languages instructions could help models gain general instruction following (chatting) ability.

**Training Dataset.** We employ LIMA (Zhou et al., 2023), a high-quality instruction tuning dataset of 1000 carefully created (instruction, answer) pairs. Furthermore, we leverage our proposed unnatural languages searching approach to find an unnatural version for each instruction in LIMA, and keep the original answers in natural version.

**Benchmarks.** We evaluate all variants on Length-controlled (LC) AlpacaEval 2.0 (Li et al., 2023) and MixEval (Ni et al., 2024). LC AlpacaEval 2.0 is a well-recognized benchmark for chat model evaluation. MixEval is a ground-truth-based benchmark that collects data from numerous QA datasets under real-world data distribution.

Table 5: The accuracy of the unnatural GSM8K test set for models instruction-tuned on various types of GSM8K training sets. Columns denote the training set.

| Unnatural Test Acc. | Natural | Random | Empty | Unnatural |
|---|---|---|---|---|
| Mistral-7B | 0.187 | 0.116 | 0.122 | **0.310** |
| Llama-3-8B | 0.197 | 0.087 | 0.109 | **0.312** |
| Mistral-7B-Inst | 0.225 | 0.138 | 0.171 | **0.300** |
| Llama-3-8B-Inst | 0.214 | 0.136 | 0.179 | **0.349** |
| *Average* | 0.206 | 0.119 | 0.145 | **0.318** |

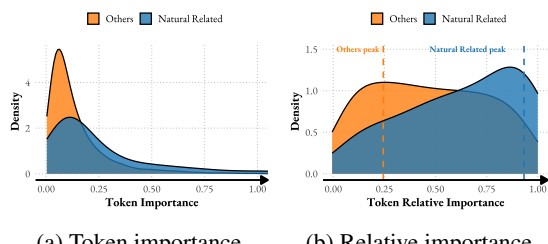

(a) Token importance  (b) Relative importance

Figure 3: Token importance of SimGSM8K.

**Baselines.** We employ three baselines: (i) Natural, which uses the original unmodified instructions; (ii) Random, which replace the original instructions with an equal number of random tokens; and (iii) Empty, in which instruction is empty (Hewitt et al., 2024).

**Experiment Details.** In practice, since the instruction in LIMA is extremely long, which exceeds the capacity of our searching approach, we leverage GPT-4 to generate a compressed version of the instructions. Therefore, for fairness, we compare the instruction following ability of models finetuned on the unnatural LIMA and the instruction-shortened LIMA. In addition, all models are fine-tuned for 10 epochs using identical hyperparameters.

## 4.2 Main Results

In Figure 2, we show the winrate of different variants against the corresponding models tuned on natural LIMA using the official pipeline and the annotation model is GPT-4o. The results clearly show that responses of models instruction tuned on unnatural LIMA is comparable to the one tuned on natural LIMA with a winrate of $48.82\%$ in average, which outperforms the baselines (models tuned on random/empty LIMA) by a large margin. Furthermore, Table 3 confirms our conclusion. As shown in Table 3, most models instruction tuned on unnatural version LIMA performs on-par with the one tuned on natural LIMA. Meanwhile, it outperforms the baselines, which are tuned on random and empty instruction version LIMA, significantly, especially under base model Llama-3-70B with a margin of over $11\%$. Both results strongly demonstrate our claim that unnatural languages contains generalizable patterns that could be help LLMs gain instruction-following ability.

## 4.3 Further Analysis

So far we have shown that unnatural languages contains natural patterns that could be generalized to various tasks by instruction tuning. Besides, we are curious about whether the unnatural languages consists unnatural patterns and whether fine-tuning on unnatural languages could boost the unnatural language understanding capability of LLMs. To this end, we focus the math reasoning task, which is highly demanded for question understanding. We created an unnatural languages version of GSM8K for its training subset and test subset. Due to the high cost of GCG and computation limitation, we searched 1333 training instances and 654 test instances. For training, we leverage corresponding answer augmentation version (i.e., for each question, there are multiple version of correct chain-of-thought answers.) from Yu et al. (2023), which finally results in 14886 training instances. Built upon the training set, we create four versions, the same as Section 4.1. We train the pre-trained version and instruction tuned version of Mistral-7B-v0.1 and Llama-3-8B on the four types of training set and test their performance on the unnatural test set. The results are shown in Table 5.

It shows that models fine-tuned on the unnatural training set significantly outperform models trained on other types of training sets when evaluated on the unnatural test set. Specifically, the average accuracy for models tuned on unnatural training set is $31.8\%$, outperforming the one tuned on natural training set by $11.2\%$, the one tuned on random training set by $19.9\%$, and the one tuned on empty training set by $17.3\%$. This indicates that unnatural languages contain generalizable unnatural patterns that could enhance LLMs' unnatural languages understanding capability.

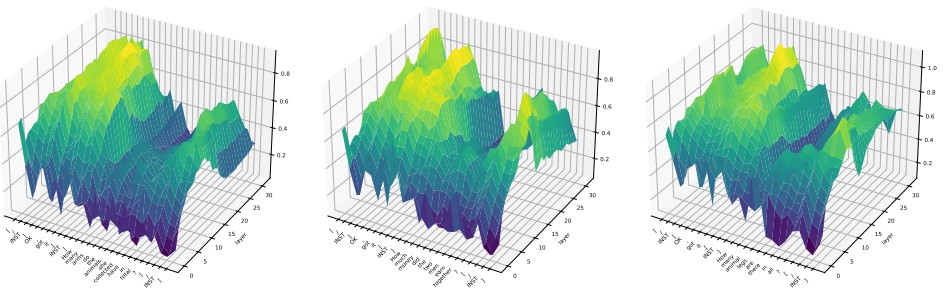

Figure 4: Three 3D surface examples showing the inverse similarity of natural and unnatural context embeddings. The inverse similarity decreases significantly when question-related tokens are added, indicating that LLMs correctly infers the organization of keywords.

## 5 How do LLMs Understand Unnatural Languages?

Unnatural languages have been empirically shown to contain latent features that are comprehensible across different LLMs, while also enhancing their ability to follow instructions. In this section, we investigate how LLMs process and understand these unnatural languages.

### 5.1 LLMs Extract Keywords from Unnatural Languages

To investigate what LLMs truly capture when processing unnatural languages, we evaluate each token's importance by measuring its impact on the output when removed from the sequence. Formally, for an unnatural string $S'$, tokenized by model $M$ as $\mathbf{x}_{1:n} = [x_1, x_2, \cdots, x_n]$, the importance of token $x_i$ is defined as the effect of its removal on the change in the embedding of $M$'s final layer, i.e.,

$$I(x_i) = \left\| \mathcal{LLM}_M(\mathbf{x}_{1:n}) - \mathcal{LLM}_M(\mathbf{x}_{1:n} \backslash x_i) \right\|_2. \tag{4}$$

Specifically, to ensure consistent evaluation across unnatural strings of varying lengths and given that the embedding of the final position is used to predict the next token, we measure the embedding of the final position rather than the entire sequence. Furthermore, we normalize the "token importance" relative to the most important token in each sequence, referring to this as "relative token importance". The relative token importance represents a softened position of the token after sorting in ascending order, scaled within the range $[0, 1]$. Tokens with greater importance within a data point have a relative position closer to $1$, whereas less important tokens have a relative position closer to $0$.

In Figure 3, we present the distribution of token importance and relative token importance within the unnatural version of the SimGSM8K dataset as processed by the model Llama-3-8B-Instruct. Specifically, as shown in the Figure 3 (a), the density of natural-related tokens (tokens appears in the natural version) is more higher than others when token importance is higher than 0.2. Besides, compared with natural related tokens, the majority of other tokens lies in the lower importance range. Furthermore, Figure 3 (b) clearly shows that most of the naturally related tokens are higher relative important while the other tokens are lower relative important. The above results demonstrate that LLMs are capable of pay more attention on the natural related tokens while filtering out the other noise. Consequently, LLMs effectively extract keywords from unnatural languages inputs.

### 5.2 LLMs Infer Correct Organization of Keywords in Unnatural Languages

Although LLMs are capable of extracting keywords from unnatural languages, the extracted words are often shuffled and arranged in the wrong order. Therefore, we hypothesize that LLMs can reorganize these keywords and infer their correct arrangement. Specifically, when the unnatural context is provided to LLMs alongside appended natural questions, we propose that LLMs progressively infer the correct organization of keywords—i.e., their corresponding natural language versions—as they process an increasing number of natural question tokens.

To verify our hypothesis, we calculate the inverse similarity of embeddings between the context inputs of unnatural languages and their corresponding natural versions across layers, as well as for

Table 6: Concrete examples of token reordering: The "natural context" represents the original version, while the "unnatural version" simplifies the unnatural languages by removing noise and retaining only keywords. We decode the internal embeddings of LLMs into tokens for unnatural language inputs using the same decoder as the final output layer, referred to as "decode internal embeddings".

| Natural Context | Unnatural Version (De-noised) | Decode Internal Embeddings |
|---|---|---|
| Brandon sold 86 geckos last year. He sold twice that many the year before. | twice geckos year before last sold Brandon He | twice geckos year before last before sold Brandon He sold he 86 sold twice sold twice sold last twice year sold |
| Ruiz receives a monthly salary of $500. | a monthly $500 salary Ruiz receives | a monthly $500 a salary a Ruiz a receives a receives a salary |

increasingly natural question tokens inputted into LLMs. As shown in Figure 4, for the marginal version of a layer, the inverse similarity of unnatural and natural context embeddings gradually decreases as following tokens are inputted. Particularly, the inverse similarity drastically decreases when the question related tokens are inputted. This indicates that LLMs does not always comprehend unnatural languages independently as the corresponding natural language. In contrast, the comprehension process is highly related to the context (i.e. the questions.). As a result, the unnatural languages works in certain contexts. Supposing random context is provided, the model behavior on unnatural languages could be different from natural language.

## 5.3 QUALITATIVE ANALYSIS

We further investigate whether LLMs are truly capable of reordering keywords by analyzing the embeddings of intermediate layers. Specifically, we decode the internal embeddings of LLMs when processing unnatural language inputs into tokens using the same decoder as the final layer. As shown in Table 6, we observe that although the keywords in the unnatural version are disordered, LLMs are able to reorder certain patterns in the keywords to match the original natural context. Moreover, we utilize dependency parsing to demonstrate that LLMs can understand the dependency structure of unnatural languages. Details are illustrated in Appendix A7.

## 6 RELATED WORKS

**Unnatural Languages.** Prior studies have observed isolated instances of unexpected model behavior, while they did not explicitly identify or systematically analyze unnatural language. For example, Zou et al. (2023) prompted models generating harmful outputs, Pfau et al. (2024) enhanced chain-of-thought reasoning, and Sinha et al. (2020) showed NLI models worked with permuted inputs. Kervadec et al. (2023) demonstrated that LLMs interpret unnatural languages differently, while Kallini et al. (2024) categorized them by perplexity, showing LLMs struggled to learn them.

**Discrete Optimization.** Prompt optimization tools have been used to explore unnatural languages in token space. Early works (Zou et al., 2023; Liu et al., 2023; Zhao et al., 2024; Chao et al., 2023; Andriushchenko, 2022) focused on adversarial prompts to jailbreak LLMs, while others (Shin et al., 2020; Jones et al., 2023) optimized prompts for specific outputs. These studies did not address whether such prompts reflect natural language features. This work investigates this question.

**Transferability of Adversarial Examples.** Unnatural languages often transfer across LLMs, similar to adversarial examples in computer vision Szegedy et al. (2014); Papernot et al. (2016); Nguyen et al. (2015). For example, Wallace et al. (2019) showed that prompts from GPT-2 transferred to larger models, and Jones et al. (2023) found that toxic prompts from GPT-2 affected davinci-002. Building on these findings, this work examines how LLMs rely on fragile, unnatural features in token space (Ilyas et al., 2019).

## 7 CONCLUSION

Our study reveals that LLMs possess the ability to comprehend unnatural languages, an incomprehensible data pattern that could convey information across models. Through systematic analysis and experiments, we demonstrate that unnatural languages contains generalizable patterns across a wide variety of LLMs, despite these models being predominantly aligned with human data. We show that models fine-tuned on unnatural instructions achieves on-par performance of instruction-following ability with models fine-tuned on natural versions. Furthermore, through comprehensive analysis, we demonstrate that LLMs process unnatural languages by filtering noise and inferring contextual meaning from filtered words.

## ACKNOWLEDGMENTS

This research is partially supported by the National Research Foundation Singapore under the AI Singapore Programme (AISG Award No: AISG2-TC-2023-010-SGIL) and the Singapore Ministry of Education Academic Research Fund Tier 1 (Award No: T1 251RES2207). We gratefully acknowledge the computing resources provided by Sea AI Lab, which significantly supported this work.

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

## LIMITATIONS AND FUTURE WORKS

In this work, we design a type of *unnatural language* that is comprehensible to LLMs and demonstrate that it contains useful features that could facilitate instruction tuning and even achieves on-par performance compared with the corresponding natural language. Essentially, the unnatural language searching is a process that increases the entropy of language (via replacing tokens in GCG) while trying to keep the semantic meaning of the original language. Therefore, the result inevitably contains the tokens that appeared in the original natural string, which is not 100% unnatural ideally. We have also performed ablation study by eliminating the original tokens from the candidates and initialize the searching control without natural tokens but found that the performance is significantly sub-optimal. We admit the limitation of current searching methods leveraging GCG and conjecture that the unnaturalness and performance for unnatural language could be further improved once there is more effective and efficient discrete space optimization approach.

Besides, the efficiency for GCG-like searching methods is limited and it is expensive to search large scale unnatural language version examples. This limits us from performing more comprehensive and large scale experiments. Once the searching expense is mitigated, unnatural language could be further explored for practical usage.

In addition, we found that the unnatural language are not always generalizable across different tasks. For example, fine-tuning on the unnatural version of GSM8K training set as in Sec. 4.3 does not achieve on-par performance with the one fine-tuned on natural set in expectation. We suspect that this is because that GSM8K questions are too complex (long) for GCG to search a compatible unnatural version. This also explains why the unnatural SimGSM8K in Sec. 3 only achieves 54% accuracy in average across models.

## A1 ALGORITHM DETAILS

Algorithm 1 presents the detailed implementation of the unnatural languages searching algorithm.

---

**Algorithm 1** Unnatural Languages Searching

---

**Input:** Natural string $S$, searching models $\mathcal{M}$ and tasks $\mathcal{T}$, batch size $B$, $k$, number of iterations $T$
1: // Initialize $x$ via shuffle words in $S$ and inject special characters.
2: **Initialization:** $x = \text{random\_inject}(\text{shuffle}(S))$
3: **repeat**
4:    **for** $M \in \mathcal{M}$ **do**
5:       // Tokenize $x$ through model $M$.
6:       $\mathbf{x}_{1:n} = \text{Tokenize}_M(x)$
7:       // Obtain top-k alternative tokens of each position in $\mathbf{x}_{1:n}$.
8:       $\mathbf{X}_{1:n} = \text{Top-k}\big(\nabla_{\mathbf{x}_{1:n}} \sum_{t \in \mathcal{T}} \log P_M(S|\mathbf{x}_{1:n}, t)\big)$
9:       **for** $b = 1, ..., B$ **do**
10:          // Uniformly sample candidates.
11:          $\tilde{\mathbf{x}}_{1:n}^{(b)} = \mathbf{x}_{1:n}$
12:          $\tilde{\mathbf{x}}_{1:n}^{(b)}[i] = \text{Uni}(\mathbf{X}_{1:n}[i]),\ i = \text{Uni}([1:n])$
13:          // Decode tokens back string.
14:          $\tilde{x}^{(b)} = \text{Decode}_M(\tilde{\mathbf{x}}_{1:n}^{(b)})$
15:       **end for**
16:    **end for**
17:    // Select the best candidate.
18:    $\tilde{x}^{b^*} = \text{argmax}_b \sum_{M \in \mathcal{M}} \sum_{t \in \mathcal{T}} \log P_M(S|\tilde{x}^{(b)}, t)$
19:    // Replace the original string with the modified string.
20:    $x = \tilde{x}^{b^*}$
21: **until** Repeat for $T$ times
**Output:** Equivalent unnatural string $S'$

---

## A2 ALGORITHM VERIFICAITON EXPERIMENTS

Here we perform a verification experiment to show that we searched unnatural strings could be translated back to natural version. Specifically, we perform such translation task by appending a translation task description 'Translate the above sentences into natural languages'. The translation performance is shown in Table A7. *EM* denotes exact match; *F1* denotes F1 score; and *NLI* measures the semantic relationship between sentences (e.g., entailment, neutral, contradiction) using pre-trained models, offering a nuanced evaluation of meaning similarity. We treat 'entailment' as positive cases and compute the accuracy to obtain the final score.

Table A7: Results of unnatural language to natural language translation task.

| Dataset | EM | F1 | NLI |
|---|---|---|---|
| SynContextQA | 0.5600 | 0.854 | 0.860 |
| SimGSM8K | 0.660 | 0.805 | 0.650 |

## A3 SYNCONTEXTQA AND SIMGSM8K DATASET DETAILS

## A4 SYNCONTEXTQA

**Generation Details.** We leverage GPT-3.5 (OpenAI, 2023) to generate context about non-existing entities and their corresponding questions. The prompt is provided as follows in Table A8. To ensure the diversity of the generated context, we generate $1,000$ candidates in total and perform k-means clustering according to the embeddings generated by a SOTA text embedding model to form 100 clusters. Finally, we select 100 instances that are closest to each of the cluster centers.

Table A8: Prompt of SynContextQA generation.

Please generate 10 synthetic business or personal case for reading comprehension. The context information should be specific to a synthetic object, e.g., 'The company TechDouDou raised 1,000,000 fundings in Q4, 2023' instead of 'A company raised 1,000,000 fundings in Q4, 2023'. all data should be different from each other as much as possible. The case contains three parts: (1) context that provides specific information, where the length should be no longer than 40 characters; (2) a question that asked about that information; (3) the corresponding answer.
[Context]:The revenue of the company Countingstar for Q1 is 100,000$.
[Question]: What is the revenue of Countingstar for Q1?
[Answer]: The revenue of Countingstar for Q1 is 100,000$

**Dataset Details.** The dataset is generated by GPT-3.5. Each data contains a simple and synthetic context related to unexisted business or personal as well as a couple of questions related to the context. Genearally, one question asks about the action of the entity while the other question asks about the name of the enitity. For example, the context is *"EcoGardens launched a new sustainable packing initiative"* and the two questions are *"What recent initiative did EcoGradens launch?"* and *"Which company launched a new sustainable packing initiative?"*. To ensure the diversity of the generated questions, we generated 1,000 contexts and leverage k-means to get 100 data points, each of which is closed to the centers of a cluster. the cluster embeddings were generated by a SOTA embedding model in `sentence-transformers` (Reimers & Gurevych, 2019). For each data point, we manually create the correct answer candidates for each question.

## A5 SIMGSM8K

**Dataset Details.** The dataset serves as a more challenging dataset compared to SynContextQA, where the data points are derived from the test set of `GSM8K`. As a result, the context is more complex which often contains multiple entities and much more information. Meanwhile, the answer of the

Table A9: Performance comparison for different contexts across different models on SynContextQA and SimGSM8K datasets under *tow-turn dialogue setting*. All answers were generated under zero-shot setting without sampling. "Direct" refers to models used for unnatural language searching, while "Transfer" indicates the implementation of searched unnatural languages.

| | Model | SynContextQA | | | SimGSM8K | | |
|---|---|---|---|---|---|---|---|
| | | Natural | Shuf-InJ | Unnatural | Natural | Shuf-InJ | Unnatural |
| **Direct** | Mistral-7B-Instruct | 0.92 | 0.47 | 0.92 | 0.85 | 0.20 | 0.42 |
| | Vicuna-7B | 0.94 | 0.49 | 0.90 | 0.63 | 0.18 | 0.21 |
| | Average | 0.93 | 0.48 | 0.91 | 0.74 | 0.19 | 0.32 |
| **Transfer** | Meta-Llama-3-8B-Instruct | 0.98 | 0.51 | 0.84 | 0.77 | 0.31 | 0.40 |
| | Gemma-2-9B-Instruct | 0.96 | 0.46 | 0.70 | 0.97 | 0.22 | 0.45 |
| | Meta-Llama-3-70B-Instruct | 0.98 | 0.70 | 0.92 | 1.00 | 0.41 | 0.73 |
| | GPT-3.5 | 0.98 | 0.68 | 0.92 | 0.92 | 0.38 | 0.49 |
| | GPT-4 | 0.99 | 0.64 | 0.91 | 0.96 | 0.32 | 0.48 |
| | Average | 0.98 | 0.60 | 0.86 | 0.92 | 0.33 | 0.51 |

question requires several steps of reasoning and the correct answer could not be found in the context directly using simple keyword matching. Since our goal is to evaluate the ability of unnatural language comprehension of LLMs, we do not want to introduce too complex QA paris which could not even be answered correctly under natural language. To this end, we selected 100 questions from the original GSM8K test set considering the context length and correctness of models.

## A6 UNNATURAL QA EXPERIMENTS UNDER TWO-TURN DIALOGUE SETTING.

To verify the model's understanding of unnatural context and given that these are chat-based models, we implement a dialogue format for context-based question-answering. Specifically, each QA session consists of two turn. In the first turn, we provide the context in unnatural language to the model, which responds with "OK, got it." In the second turn, we pose the question related to the previously provided context and evaluate the model's response accuracy through keyword exact matching. Detailed results are shown in Table A9. The results serves as a complimentary results of Table 2 and it further demonstrates that such unnatural languages is highly transferrable across models.

## A7 LLMS UNDERSTAND THE DEPENDENCY STRUCTURE OF UNNATURAL LANGUAGE.

Dependency parsing is one of the most commonly used techniques to analyze the syntactic structure of natural sentences. Dependency parsing transfer a sentence into a tree, where each word/token is a node and the directed edges represent the dependency, the end node (child) depends on, e.g. modifying or being arguments of, the source node (parent). In general, a fake ROOT node is usually added to the whole tree such that every actual word have parents. Dependency parsing serves as a fundamental technique for understanding the syntax of sentences. In this section, we are curious about how LLMs understand the unnatural language. A more specific question is that how LLMs interpret the syntax of unnatural language since they could understand it.

Recently, Hewitt & Manning (2019) showed that pretrained LMs', e.g. BERT (Devlin, 2018), output embeddings contains the structural syntax information of the inputs. One can simply leverage the probing (a linear transformation) techiques to extract such information and build a dependency syntax tree. Leveraging a modified version of the so-called *structural probing*, we train a linear head upon a freezed pre-trained model (e.g. `Llama-3-8B`) to predict the dependency syntax tree of the input sentence using natural language corpus as training source. Then we leverage the trained head to predict the syntax tree of unnatural sentences. We introduce the structural probing method as follows.

A syntax tree of a sentence $l$ is equal to a directed acyclic graph (DAG) $G = (V, E)$, where $V$ is the set of words and $E$ denotes the set of edges. to build the guidance for training, we compute a

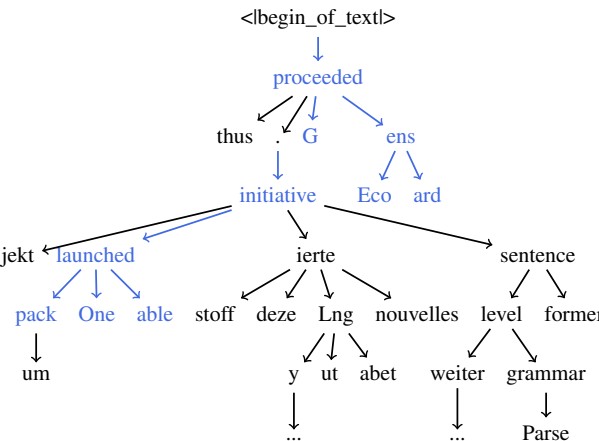

Figure A5: A dependency syntax tree of unnatural sentence: "| *EcoGardenslaz proceeded thus,- sust "able deix um nouvelles packstoff launchedierteutabetLng initiative. Onejekt y deze sentence former GETTRAN()->()} grammar level:: EGB »OK!. Parse Sie fast{-itt weiter*", whose corresponding natural version is "*EcoGardens launched a new sustainable packaging initiative*". The tree was generated by `Llama-3-8B` via structural probing (Hewitt & Manning, 2019). We annotate a sub-tree which reasonably represents the semantic meaning of "*EcoGardens proceeded initiative that launched one sustainable pack.*"

distance matrix $\mathbf{d} \in \mathbb{N}^{(|V|,|V|)}$ for the graph. The element $d_{ij}$ of the distance matrix is defined as the length of the shortest path between node $i$ and node $j$. Particularly, if two nodes are adjacent, the distance is 1. In the original paper, the authors omitted the root node for syntax tree when building the distance matrix, which results in a symmetric matrix that loses the information of root node and edge directions. In our work, we treat the root node as a concrete token (the BOS special token for any LLM tokenizer) and build the corresponding distance matrix. Such distance matrix is an one-to-one mapping of the syntax tree since one can easily recover the syntax tree since the root node is known and the tree is acyclic. In the following parts, we use $\mathbf{d}^l \in \mathbb{N}^{(|V_l|+1,|V_l|+1)}$ denoting the distance matrix for sentence $l$.

Given a model's output embedding $\mathbf{h}^l \in \mathbb{R}^{(L,H)}$ for sentence $l$. We formalize the training process as the following optimization problem:

$$\min_{\mathbf{W}} \sum_l \frac{1}{|s^l|^2} \sum_{i,j} |\mathbf{d} - f_{\mathbf{W}}(\mathbf{h}_i^l, \mathbf{h}_j^l)|, \tag{5}$$

where $f_{\mathbf{W}} = (\mathbf{W}(\mathbf{h}_i^l - \mathbf{h}_j^l))^T (\mathbf{W}(\mathbf{h}_i^l - \mathbf{h}_j^l))$ computes the squared eucliean distance between two nodes. $|s^l|$ denotes the number of tokens in sentence $l$. In practice we train the probing head with the data from EN_EWT training set of universal dependency (Silveira et al., 2014).

Then we use the trained probing head to predict the dependency syntax tree of unnatural sentences. We show an example in Figure A5. As shown in Figure A5, the dependency tree clearly contains the syntax structure whose semantic meaning is similar to the natural language. This indicates that LLMs could capture the syntax structure of unnatural language that contains information akin to the natural language.

