# OpenReview forum: "Unnatural Languages Are Not Bugs but Features for LLMs"
_ICLR.cc/2025/Workshop/BuildingTrust — BuildingTrust_

### Official Review · Reviewer_bkas · 2025-02-19
**Good paper**

**Rating:** 7
**Confidence:** 3

**Review:**

Strengths of the Paper：
Strong Empirical Evidence and Benchmarks

The authors present compelling empirical results showing that unnatural language constructs retain semantic meaning and can generalize across different models.
The paper evaluates models on SynContextQA and SimGSM8K, carefully designed to assess whether LLMs can extract meaning from unnatural contexts.
The inclusion of a two-turn dialogue format strengthens the claim that LLMs genuinely process unnatural text.

Methodological Rigor in Finding Unnatural Representations

The paper introduces a structured search method to identify unnatural versions of text using a gradient-based stochastic sampling approach despite deriving from the GCG.
A key strength is the optimization process across multiple LLMs, ensuring that discovered unnatural language patterns are not overfitted to a single model.


Transferability of Unnatural Language Representations

The study finds that models trained on unnatural language instructions perform on par with models trained on natural language.
This suggests that unnatural language constructs contain latent features that support task generalization

Weaknesses:

Lack of Human Interpretability

While the paper claims that unnatural languages retain latent meaning for LLMs, it does not sufficiently explore whether such representations align with human cognition.
It would be valuable to conduct human annotation studies to assess if these representations are systematically interpretable.

Limited Scope of Tasks Considered

The evaluation focuses on QA and instruction tuning tasks. However, task complexity varies widely across NLP domains, and it remains unclear whether unnatural languages hold their generalization properties in reasoning-intensive tasks or low-resource languages.

---

### Official Review · Reviewer_1NTv · 2025-03-02
**A good paper with well conducted experiments and relevant findings.**

**Rating:** 8
**Confidence:** 3

**Review:**

### Summary
The paper systematically investigates whether unnatural languages—strings that appear incomprehensible to humans but retain semantic meaning for LLMs—contain latent features that can transfer between LLMs. The authors propose a search technique to generate unnatural strings from natural text and use it to analyze LLM performance. They compare LLM performance on tasks with unnatural context versus natural context and demonstrate that LLMs can learn from instructions in an unnatural form.

### Strengths
- The paper provides a systematic analysis of how unnatural languages influence LLM performance and investigates LLMs' ability to learn from unnatural instructions. To the best of my knowledge, such a study has not been conducted before—only isolated cases of unnatural languages causing surprising behaviour in LLMs have been observed.
- The study examines the transferability of these methods across different, recent LLMs, making the research more comprehensive.
- The experiments are well-designed with appropriate counterfactuals.
- The authors analyse how LLMs process unnatural languages, showing that LLMs filter relevant words. This claim is supported by multiple experiments.

### Weaknesses
- It is not discussed (or I missed it) why in some cases the performance with the unnatural languages is better if the models are processing them by extracting only relevant keywords which seems to be harder task than directly processing words in the right order (as in natural languages).
- In Table 5, the reasoning behind bolding certain results appears inconsistent.

Overall, I think this is a good paper with well conducted experiments and relevant findings.

---

### Official Review · Reviewer_7piL · 2025-03-03
**Interesting findings. Would be helpful to add non-finetuned baselines to the tables and winrate comparisons.**

**Rating:** 7
**Confidence:** 3

**Review:**

### Summary
Investigates an interesting observation that unnatural languages generalize between different models and model families, indicating that they must have some generalizable meaning to LLMs.

### Strengths:
- Clearly written
- An interesting finding and lots of thoughtful investigation.

### Weaknesses/possible improvements:
- A table for the AlpacaEval results would be helpful.
- For the AlpacaEval win rate, it is more meaningful to look at win rate (base model vs. natural-finetuned) and (base model vs. unnatural-finetuned), rather than just (natural-finetuned vs. unnatural-finetuned) since a win rate of 50% (as in the current results) could be achieved by comparing two arbitrarily bad models.
- Similarly, it would be helpful to add another column to Table 4 for the base model with no finetuning.
- The sentence on line 234 is missing some words?
- The acronym GCG for greedy coordinate gradient (?) is used without being introduced anywhere.
- I would be interested to know whether numbers are ever/often changed in the unnatural versions, since, for example, in gsm8k these are the important tokens.
- In Table 5, I wonder to what extent the unnatural-finetuning is just teaching the model that when the prompt looks unnatural it should just output a number (or even a guess at a number based on the numbers in the prompt). The natural-finetuned model would not have this bias towards guessing numbers when the prompt is unnatural, and so this could explain the increase in performance. There may be some ablations that could help disentangle this.

---

### Decision · Program_Chairs · 2025-03-04

**Decision:**

Accept

**Comment:**

This paper presents experiments demonstrating that unnatural language constructs retain semantic meaning and further this generalizes across different models.